# Beyond Green: The Therapeutic Potential of Chlorophyll and Its Derivatives in Diabetes Control

**DOI:** 10.3390/nu17162653

**Published:** 2025-08-15

**Authors:** Giovanni Sartore, Giuseppe Zagotto, Eugenio Ragazzi

**Affiliations:** 1Department of Medicine—DIMED, University of Padova, 35128 Padova, Italy; g.sartore@unipd.it; 2Department of Pharmaceutical and Pharmacological Sciences, University of Padova, 35131 Padova, Italy; giuseppe.zagotto@unipd.it; 3Studium Patavinum, University of Padova, 35122 Padova, Italy

**Keywords:** chlorophyll, pheophorbide *a*, pheophytin *a*, chlorophyllin, diabetes, α-glucosidase inhibition

## Abstract

Chlorophyll, the green pigment essential for photosynthesis, abundantly found in green vegetables and algae, has attracted growing scientific interest for its potential therapeutic effects, particularly in diabetes management. Recent research highlighted that chlorophyll and its derivatives may beneficially influence glucose metabolism and oxidative stress, key factors in diabetes. This review examines current knowledge on how chlorophyll compounds could aid diabetes control. Chlorophyll and its derivatives appear to support glucose regulation primarily through actions in the gastrointestinal tract. They modulate gut microbiota, improve glucose tolerance, reduce inflammation, and alleviate obesity-related markers. While chlorophyll itself does not directly inhibit digestive enzymes like α-glucosidase, its derivatives such as pheophorbide *a*, pheophytin *a*, and pyropheophytin *a* may slow carbohydrate digestion, acting as α-amylase and α-glucosidase inhibitors, reducing postprandial glucose spikes. Additionally, chlorophyll enhances resistant starch content, further controlling glucose absorption. Beyond digestion, chlorophyll derivatives show promise in inhibiting glycation processes, improving insulin sensitivity through nuclear receptor modulation, and lowering oxidative stress. However, some compounds pose risks due to photosensitizing effects and toxicity, warranting careful consideration. Chlorophyllin, a stable semi-synthetic derivative, also shows potential in improving glucose and lipid metabolism. Notably, pheophorbide *a* demonstrates insulin-mimetic activity by stimulating glucose uptake via glucose transporters, offering a novel therapeutic avenue. Overall, the antioxidant, anti-inflammatory, and insulin-mimicking properties of chlorophyll derivatives suggest a multifaceted approach to diabetes management. While promising, these findings require further clinical validation to establish effective therapeutic applications.

## 1. Introduction

Diabetes mellitus is a chronic metabolic disorder characterized by hyperglycaemia resulting from defects in insulin secretion, insulin action, or both. It primarily exists in two major forms: type 1 diabetes (T1D), an autoimmune condition leading to the destruction of pancreatic β-cells, which are responsible for the production and secretion of insulin; and type 2 diabetes (T2D), which involves insulin resistance and progressive β-cell dysfunction [1]. The global burden of diabetes continues to grow, with 589 million adults aged 20–79 affected worldwide—a number projected to rise significantly in the coming decades [2].

The therapeutic approaches to T1D and T2D differ significantly due to their distinct pathophysiological mechanisms [1]. Insulin replacement therapy remains the cornerstone of T1D management. Modern regimens often include multiple daily injections using basal-bolus strategies or continuous subcutaneous insulin infusion via insulin pumps. Recent advancements include ultra-rapid insulin analogues, smart pens, and hybrid closed-loop systems that automate insulin delivery based on continuous glucose monitoring data [3,4]. Also, promising new immunotherapies have emerged and are being developed [5].

Treatment of T2D typically begins with lifestyle modification—including diet, physical activity, and weight management—and progresses to pharmacologic therapy if glycaemic goals are not achieved. First-line pharmacotherapy commonly involves metformin, due to its efficacy, safety, and low cost [6,7,8]. In patients with established cardiovascular disease or high-risk features, sodium-glucose transport protein 2 (SGLT2) inhibitors and glucagon-like peptide 1 (GLP-1) receptor agonists are increasingly recommended for their dual glycaemic and cardiovascular benefits [6]. Other options include dipeptidyl peptidase 4 (DPP-4) inhibitors, thiazolidinediones, sulfonylureas, and, in some cases, insulin. While insulin is essential in T1D, it may only be introduced later in the treatment of T2D, particularly when oral agents are no longer sufficient to maintain adequate glycaemic control [6,7].

Alternative and complementary therapies have been proposed in the management of diabetes by lowering blood glucose [9]. There is great interest in the use of plant and plant-derived substances for effective treatment of diabetes and its complications [10,11,12,13,14], including cardiac disorders [15]. For instance, several herbs have been demonstrated to reduce glucose absorption by inhibiting α-glucosidase [16], or by improving sensitivity for endogenous insulin, as well as protecting pancreatic β-cells from damage [10], and providing useful antioxidant nutraceuticals [17] to counteract the deleterious effect of advanced glycation end-products.

In this context, it is well known that an adequate daily assumption of fruits and vegetables can reduce risk factors for many communicable and non-communicable diseases [18], including diabetes [19]. A useful tip to assure adequate intake of fruits and vegetables and optimize the assumption of useful protective substances is the colour: “eating a rainbow” of fruits and vegetables is a common suggestion to ensure a wide range of healthy nutraceuticals, as documented by a recent umbrella review that provided evidence of healthy effects of colour-associated bioactive pigments [18]. Although the health benefits of variously coloured pigments—such as lycopene (red), alpha- and beta-carotene (yellow and orange), anthocyanins (red, purple, and blue), flavonols (pale yellow), and flavones (white)—are well-documented [18], the role of the common and ubiquitous chlorophyll (green) remains comparatively underexplored. Even after more than two centuries of research, chlorophyll continues to be a subject of active investigation and discussion, as evidenced by recent studies and publications [20]. Chlorophylls are green pigments of plants that play a central role in photosynthesis and have recently gained attention for their potential therapeutic properties, including antioxidant, anti-inflammatory, anti-obesogenic and antidiabetic effects [21]. Anecdotal reports also refer to the use of chlorophyll in treating chronic ulcers as a complication of diabetes [22,23]. More recently, chlorophyll catabolites have been proposed as potent potential candidates for antiviral therapy against SARS-CoV-2 [24]. In the context of diabetes management, emerging studies suggest that chlorophylls and their derivatives may influence glucose metabolism and oxidative stress pathways.

Interestingly, a historical and complementary perspective from chromotherapy associates the colour green with balance and healing. According to Pandey et al. [9], chromotherapy considers diabetes to be linked to a deficiency of orange, yellow, and green colours in the body’s energy system, and green light is believed to support internal harmony. While this concept is anecdotal and lacks scientific validation, the alignment between the green colour of chlorophyll and its potential biological effects offers a curious, though speculative, intersection of traditional colour-based theories and modern biomedical research.

In light of the above considerations, the aim of the present review is to evaluate the current knowledge on the potential effects of chlorophyll and its derivatives in the context of diabetes, including their mechanisms of action, preclinical and clinical evidence, and directions for future research.

## 2. Literature Search Strategy

Although this is a narrative review, a structured approach was adopted for article selection. A comprehensive literature search was conducted using PubMed, Scopus and Google Scholar to identify relevant studies on chlorophyll and its derivatives in the context of diabetes. The search in the PubMed and Scopus databases employed the terms: (chlorophyll AND diabetes) OR (chlorophyll derivatives AND diabetes) OR (chlorophyllin AND diabetes) OR (pheophorbide AND diabetes) OR (pheophytin AND diabetes), with no time restrictions applied. Similarly, Google Scholar searches used key phrases like “chlorophyll AND diabetes”, “chlorophyllin AND diabetes”, and “chlorophyll derivatives AND diabetes”. Only articles published in English were considered. Studies were selected based on their relevance to the biological roles, mechanisms of action, and therapeutic potential of chlorophyll compounds in diabetes. Selected articles were then grouped thematically to highlight emerging trends and research gaps.

## 3. Chlorophylls and Related Compounds

Chlorophyll (from Greek χλωρός, green and φύλλον, leaf) is the most typical pigment present in the chloroplast grains of plant cells [25,26]. Chlorophyll *a*, *b*, *d*, and *f* have a chlorin nucleus (17,18-dihydroporphyrin) with a magnesium atom at the centre and a propionate residue with a phytyl ester. Porphyrin is a heterocycle with 22 conjugated π-electrons as shown in Figure 1, where the 18 π-electrons involved in the aromatic system are highlighted in red. The other four π-electrons possess more double-bond characteristics and are not involved in the aromatic system (Figure 1).

This conjugated macrocycle, with or without the metal atom, is primarily responsible for the colours of many biological molecules containing the porphyrin ring. Different substituents lead to variations in the electronic distribution and energy, resulting in different absorption spectra in the visible range and, consequently, different colours (e.g., haemoglobin/myoglobin and chlorophyll). Carbon atoms of the porphyrin nucleus can be chemically and structurally classified as C_a_, C_b_ and C_m_. Among these, only the C_m_ and C_b_ carbon atoms can undergo various addition and substitution reactions. Chlorophylls are magnesium (rarely zinc) complexes, where the magnesium atom is tetra-coordinated with the four porphyrinic nitrogens and has a fifth axial ligand in solution within a polypeptide matrix (Figure 2). By contrast, haeme consists of the tetrapyrrole protoporphyrin IX (Figure 3) with a single iron atom. In haemoglobin, the iron is coordinated to the four porphyrin nitrogen atoms, while one axial site is ligated to an amino acid residue that links the haeme to the protein, and the other axial site binds oxygen, carbon dioxide, or a protein amino acid.

Electrons in the porphyrin system can move freely within the ring structure, and energized electrons, generated by sunlight photons, transfer chemical energy to other substrates, enabling the synthesis of carbohydrates. Chlorophyll also presents a hydrophobic tail that permits insertion into the chloroplast thylakoid membrane, which is a specialized membrane system found inside the chloroplasts of plant cells, as well as in cyanobacteria, playing a fundamental role in photosynthesis.

Several kinds of chlorophyll exist, but the most relevant ones are chlorophyll *a* and *b* (Figure 3). Their structures differ by a single substituent on the chlorin ring: chlorophyll *a* possesses a methyl group, whereas in chlorophyll *b*, this methyl is oxidized to a formyl group. As previously mentioned, these different substituents result in distinct light absorption properties, enabling a broader range of the visible spectrum to be absorbed, primarily in the blue and red wavelengths. The absorption peaks of chlorophyll *a* are at 372 nm and 642 nm, while for chlorophyll *b* they are 392 nm and 626 nm, as determined free of perturbing solvent interactions [27].

Pheophytin is a dark bluish pigment obtained by treating chlorophyll with a weak acid, or after thermal treatment [28]; the molecule lacks the central magnesium ion (Figure 3). As with chlorophyll, two main types of pheophytin exist: pheophytin *a* and pheophytin *b*, corresponding to chlorophyll *a* and *b*, respectively.

Pheophorbide is obtained by chlorophyll breakdown [28] and consists, like pheophytin, of a chlorin ring lacking the central magnesium ion, but also without the phytyl tail (Figure 3). As with pheophytin, two main types of pheophorbide exist: pheophorbide *a* and pheophorbide *b*, corresponding to chlorophyll *a* and *b*. Interestingly, pheophorbide is structurally similar to animal protoporphyrin IX (Figure 3, inset), highlighting a conserved molecular framework that links the plant and animal kingdoms through shared tetrapyrrolic scaffolds.

Although pheophytin and pheophorbide are chlorophyll breakdown products found in plants [29], pheophytin is the predominant derivative formed during chlorophyll digestion in animals, accounting for approximately 75–77% of total chlorophyll pigments after gastrointestinal processing [30] and it also arises during thermal treatment of foods [28].

## 4. Natural Sources of Chlorophyll and Dietary Intake

Chlorophylls are widely present in many dietary sources, particularly those with dark green coloration, including leafy green vegetables, green beans, seaweed and algae [28]. Recent research underscores the nutritional value of the “Couve-Manteiga” collard green (*Brassica oleracea* var. *viridis*), particularly its high chlorophyll content and suitability for convenience food applications aimed at enhancing daily nutrient intake [31]. Average intake of chlorophylls varies based on dietary habits, especially consumption of green vegetables. By using modern mathematical techniques, a recent study has evaluated the chlorophyll intake in different European countries [32], obtaining a mean intake of 207 mg of green chlorophylls/day per adult person. Assuming all green vegetables consumed were spinach (containing 39 mg chlorophyll per gram of dry weight and 12.5% dry matter), the estimated daily intake of chlorophyll would be around 375 mg per person [28]. Chlorophyll supplements containing for instance powered chlorella algae, or chlorophyll derivatives (such as chlorophyllin, see below) are also available and used for their proposed antioxidant, anti-inflammatory, and detoxifying effects [28].

## 5. Nutritional and Functional Role of Chlorophyll

Mammalian studies reveal that the absorption of dietary chlorophyll ranges from 1–4%, while the primary route of elimination is faecal excretion of its metabolites [28,33]. Recent studies have demonstrated that natural chlorophylls are absorbed and metabolized within the body, with a tendency for pheophorbide derivatives to accumulate in the liver [21,34,35]. The liver also plays a role in converting these compounds into metabolites like phytyl-chlorin e6 [36] (Figure 4), highlighting the ongoing exploration of new chlorophyll-related compounds [28,35].

The health benefits of chlorophyll are also related to its ability to resist digestion and its absorption. Viera et al. [37] explored whether food composition influences chlorophyll bioaccessibility. Their findings showed that chlorophyll stability during digestion varies widely (15–85%) and may be influenced by salt content. Most absorbable chlorophylls in mixed micelles were pheophytins due to their high digestive stability, except in foods rich in pheophorbides. While food composition can affect digestive outcomes to some extent, chlorophyll bioaccessibility is primarily determined by its chemical structure, especially pheophorbide content. Experiments using human Caco-2 intestinal cell lines demonstrated that chlorophyll is taken up primarily in the form of its digestion-derived derivatives, including pheophytins and allomerized pheophytins, highlighting their potential bioavailability and physiological relevance [30].

## 6. Effects of Chlorophyll and Its Derivatives on Glucose Metabolism

Emerging research indicates that chlorophyll and its derivatives can influence glucose metabolism through various mechanisms in the gastrointestinal tract.

Chlorophyll supplementation has been shown to positively affect the gut microbiota composition, which is closely linked to glucose metabolism. In a study involving high-fat diet-induced obese mice [38], early-life chlorophyll supplementation (0.18 mg per 10 g body weight each day for 13 weeks) resulted in improved glucose tolerance and reduced low-grade inflammation, as well as reduced adipose tissue accumulation, counteracting obesity. This effect was associated with a significant reversal of gut dysbiosis, notably a decreased Firmicutes-to-Bacteroidetes ratio, with an increase in *Blautia* and *Bacteroidales* S24-7 group members and a reduction in *Lactococcus* and *Lactobacillus,* suggesting that chlorophyll’s impact on gut microbiota may play a role in modulating glucose absorption and metabolism, as well as favourable effects on obesity-related parameters [38].

Although chlorophyll itself has not been shown to directly inhibit human α-glucosidase enzymes, its derivatives have attracted interest for their potential metabolic effects. Alpha-glucosidase enzymes, in particular maltase-glucoamylase and sucrase-isomaltase, are located in the brush border of the small intestine and play a critical role in carbohydrate digestion by breaking down oligosaccharides into glucose [39]. The *N*-terminal (near the membrane-bound end) and *C*-terminal (luminal) subunits of both enzymes possess catalytic activity, with inhibitors—like acarbose, an oral α-glucosidase inhibitor approved by the U.S. Food and Drug Administration (FDA) for use in the management of T2D [40]—effectively binding to their highly specific, hydrophilic active sites by mimicking carbohydrate structures [39]. In contrast, chlorophyll molecules are large, lipophilic, and structurally dissimilar to native substrates, making direct interaction with α-glucosidase unlikely. However, although definitive evidence is lacking, computational docking and preliminary bioactivity studies suggest that chlorophyll derivatives warrant further investigation as modulators of carbohydrate metabolism [41].

Wang et al. [42] investigated the effects of chlorophylls and of the Mg^2+^-free derivative, pheophytin *a*, on starch digestion in vitro. The findings revealed that both chlorophylls and pheophytin *a* significantly decreased starch hydrolysis while increasing the content of resistant starch. Scanning electron microscopy demonstrated that chlorophylls either existed in free form or were absorbed and embedded on the surface of starch granules, suggesting a direct interaction that impedes enzymatic activity. By means of spectrometry and molecular docking, the authors were able to demonstrate that the phytol chain of chlorophyll forms a double helix structure with starch, hindering the accessibility of digestive enzymes such as α-amylase and α-glucosidase, thereby slowing down carbohydrate breakdown. Also, the porphyrin ring can interact with specific sites of the enzymes. By inhibiting these enzymes in two different ways, chlorophyll may lead to a more gradual glucose absorption in the gut, potentially aiding in blood glucose regulation. This mechanism makes chlorophyll and its derivatives potential natural agents for controlling postprandial blood glucose levels, offering a dietary strategy for diabetes management.

Turkiewicz et al. [43] explored the use of polysaccharide- and protein-based microencapsulation carriers to enhance the stability and bioavailability of chlorophyll-rich extracts, aiming to improve their antidiabetic potential through inhibition of digestive enzymes. This innovative encapsulation approach protected chlorophyll compounds from degradation, enabling controlled release and improved interactions with biological targets. Functionally, the microencapsulated chlorophyll extracts demonstrated significant in vitro inhibitory activity against α-amylase and α-glucosidase. While these enzymes are involved in carbohydrate digestion and glucose release, their inhibition primarily targets postprandial hyperglycaemia control. However, DPP-4 plays a distinct role by degrading incretin hormones, which regulate insulin secretion, making its inhibition a key strategy for enhancing the insulin response and prolonging glucose homeostasis [44]. In this context, Turkiewicz et al. [43] demonstrated that chlorophyll-rich microcapsules exhibited statistically significant DPP-4 inhibitory activity, with IC_50_ values ranging from 20.38 mg/mL for pea protein isolate (PPI)-based capsules to 434.63 mg/mL for maltodextrin-based capsules. Notably, soy protein isolate (SPI)-based capsules showed exceptional potency, with IC_50_ values below 0.01 mg/mL, indicating that minimal extract concentrations were sufficient to effectively suppress DPP-4 activity. The highest inhibitory potential was generally associated with formulations dried at lower temperatures and those rich in carotenoids, pheophorbides, and chlorophylls, as evidenced by strong correlations (*r* = 0.88, 0.87, and 0.79, respectively). The enhanced bioactivity of SPI- and PPI-based microcapsules was attributed to a potential synergistic effect between chlorophyll derivatives and peptides known to inhibit DPP-4. Despite these promising findings, further mechanistic studies, such as molecular docking or structural analyses, are needed to clarify the precise binding interactions of chlorophyll-related compounds with the DPP-4 active site. The combined inhibition of these enzymes suggests a multifaceted mechanism by which microencapsulated chlorophyll extracts may support glycaemic control. This research underscores the potential of advanced biopolymer carriers not only to preserve chlorophyll integrity but also to potentiate its enzymatic inhibition effects, offering a promising strategy for natural, plant-based antidiabetic therapies.

Further in vitro studies have recently provided compelling evidence that chlorophyll derivatives can directly inhibit carbohydrate-digesting enzymes, specifically α-amylase and α-glucosidase [41]. Molecular docking studies revealed that pheophytin *a* and pyropheophytin *a*, both Mg^2+^-free chlorophyll derivatives formed during digestion or heating, can effectively bind to the active site region of α-glucosidase. The docking scores indicated strong binding affinities, with pheophytin *a* and pyropheophytin *a* exhibiting binding energies of approximately −9.5 and −9.2 kcal/mol, respectively. Both compounds occupy a hydrophobic pocket near the enzyme’s catalytic site, primarily stabilized through hydrophobic interactions and hydrogen bonds. Notably, pyropheophytin *a* formed more hydrogen bonds (four compared to the two of pyropheophytin *a*) with residues adjacent to the catalytic triad, which likely underpins its relatively higher inhibitory potency. Although neither molecule directly interacted with the catalytic triad residues critical for enzymatic activity, their binding induced conformational changes in the enzyme, suggesting a non-competitive inhibition mechanism. These docking insights, corroborated by fluorescence quenching and spectroscopic analyses, highlight the potential of pheophytin *a* and pyropheophytin *a* as natural α-glucosidase inhibitors that could contribute to blood glucose regulation.

Chlorophyll derivatives such as pheophorbide *a*, which lack both the phytyl tail and central magnesium ion, exhibit increased polarity and structural flexibility. These modifications may enhance their ability to interact with metabolic enzymes, either at allosteric sites or through nonspecific binding. Supporting this hypothesis, Kim et al. [45] demonstrated in streptozotocin-induced diabetic mice that pheophorbide *a*—isolated from the edible red seaweed *Gelidium amansii*, commonly found along the coasts of Japan and Korea—showed potent inhibitory activity against α-glucosidase and α-amylase. This activity effectively reduced hyperglycaemia following a controlled feeding, better than acarbose, which was used as a positive control. In this context, acarbose, currently used clinically to manage diabetes, is widely recognized as a classical reference compound in studies on intestinal α-glucosidase inhibition [46]. The crystal structure of the *C*-terminal domain of a typical α-glucosidase, the human maltase-glucoamylase (MGAM-C), available under PDB ID as 3TOP (https://doi.org/10.2210/pdb3TOP/pdb, accessed on 12 July 2025) and 3TON (https://doi.org/10.2210/pdb3TON/pdb, accessed on 12 July 2025), can provide insights into the molecular interactions between the enzyme and various inhibitors, including acarbose [47]. Structural analysis reveals that acarbose forms multiple hydrogen bonds with key residues in the MGAM-C active catalytic domain, such as Asp1157, Asp1279, Arg1510, Asp1526, Thr1528, and His1584 [47]. These interactions play essential roles in substrate stabilization, ligand positioning, and catalysis. Additionally, Trp1355 and Phe1559 contribute through hydrophobic interactions, further anchoring the inhibitor within the binding pocket. Collectively, these contacts mimic natural substrate engagement and effectively block enzymatic activity.

As a complement to the present review, a molecular docking exploration was performed using CB-Dock2 web-based software (https://cadd.labshare.cn/cb-dock2/index.php, accessed on 12 July 2025), indicating that the chlorophyll *a* degradation product pheophorbide *a* also exhibits affinity for MGAM-C (PDB ID: 3TOP). Two potential binding cavities can be identified on chains A and B of the enzyme, with volumes of 1083 Å^3^ and 683 Å^3^, respectively, and corresponding Vina scores of −8.8 and −8.4 kcal/mol. Notably, the 683 Å^3^ pocket on chain B includes ten residues—Asp1157, Pro1159, Tyr1251,Trp1355, Trp1369, Arg1510, Asp1526, Thr1528, Phe1559 and Phe1560—that are also involved in acarbose binding (according to Ren et al. [47]), suggesting a shared interaction profile between the two ligands (Figure 5).

In addition to its inhibitory effects on digestive enzymes, pheophorbide *a*, isolated from *Capsosiphon fulvescens*, a green alga growing along the coast of Korea, Japan, North Atlantic and Pacific, and used as functional food [48], has shown promising anti-glycation activity relevant to diabetes-related complications [49]. Advanced glycation end-products (AGEs), which accumulate under hyperglycaemic conditions, contribute to oxidative stress and tissue damage in diabetic patients [50]; the study by Hong et al. [49] demonstrated that pheophorbide *a* effectively suppressed AGE formation in vitro, suggesting a protective role against glycation-mediated cellular dysfunction. This scavenging ability positions chlorophyll derivatives like pheophorbide *a* as potential multifunctional agents for mitigating both glycaemic and oxidative stress in diabetes management. Due to the known photosensitizing activity of pheophorbide *a* and pyropheophorbide *a* [51], caution is warranted in their use. Pheophorbide *a*, in particular, has been identified as a potent photosensitizer, with its accumulation linked to phototoxic skin reactions resembling pseudoporphyria, especially in individuals consuming commercial chlorophyll supplements or laver products [51,52,53,54]. Case reports have described severe photosensitivity reactions and bullous skin lesions following self-administration of chlorophyll-containing supplements [52], and a case series by Zhao et al. [53] highlighted prolonged photosensitivity persisting even after cessation of chlorophyll intake. These reactions may be due to accumulation of pheophorbide a in the skin and subsequent activation by sunlight, leading to porphyria-like symptoms. In one study, Hwang et al. [51] quantified high levels of pheophorbide *a* and pyropheophorbide *a* in dried laver implicated in food-related phototoxicity cases, underscoring the importance of analytical monitoring of commercial products. Moreover, Cinčárová et al. [54] have emphasized the need for quantitative assessment and regulatory control of pheophorbide content in algae-based supplements, as even trace amounts may pose a clinical risk when accumulated. This adverse effect may represent a significant limitation to the clinical application of these compounds, particularly in long-term or high-dose contexts. Therefore, thorough safety evaluations, including controlled dosage, proper formulation, and patient education on sun exposure, is essential before considering possible pheophorbide-based interventions in diabetes therapy. Phytanic acid, a metabolite of chlorophyll-derived phytol (Figure 6) produced by rumen microbiota and marine organisms [55,56,57], may influence metabolic regulation relevant to diabetes by acting as a potent natural agonist of retinoic X receptor (RXR) at physiological concentrations [58,59]. RXR agonists act as insulin sensitizers by activating heterodimers formed between RXR and peroxisome proliferator-activated receptor gamma (PPARγ), thereby improving glucose and lipid metabolism in models of T2D [60]. Based on this evidence, Elmazar et al. [61] conducted molecular docking simulations, which permitted the suggestion that phytanic acid is able to fit within the PPARγ ligand binding domain via four hydrogen bonds with affinities comparable to those of the insulin sensitizer thiazolidinediones. These data support the finding by Heim et al. [62], who found in rat primary hepatocytes that phytanic acid can activate PPARγ, RXR and PPARα as well, suggesting its potential role in improving insulin sensitivity and managing T2D.

Phytanic acid promotes adipocyte differentiation in both 3T3-L1 cells and human pre-adipocytes, likely by acting as a natural rexinoid (a ligand that activates RXRs), suggesting potential therapeutic value for T2D and obesity [63]. However, a 2023 review [64] questioned these findings, highlighting the limited and inconclusive evidence regarding phytanic acid’s antidiabetic effects and emphasizing the need for further research to confirm its efficacy and safety. High levels of phytanic acid can cause health risks, including neuronal damage through mitochondrial dysfunction and disrupted calcium balance, oxidative stress, and inflammation. Phytanic acid can cause vascular and skin cell toxicity and has been suggested to be linked to Refsum’s disease and other peroxisomal enzyme deficiency diseases, as well as to Alzheimer’s disease and some lymphomas. Overall, excessive phytanic acid accumulation is harmful, especially to the nervous system and through oxidative mechanisms [56].

Based on the fact that chlorophyll metabolites may act as RXR agonists, Wunderlich et al. [65] considered another, more direct, potential anti-diabetic activity. The authors investigated the effects of chlorophyll *a*, incorporated into P123 micellar copolymer, combined with photostimulation on in vitro hepatocytes from a rat model of T1D. The findings, confirmed by post-analysis in situ perfusion of intact liver, demonstrated that chlorophyll *a* decreased hepatic glucose release, stimulated glycolysis, and reduced oxidative stress markers in liver cells, indicating its potential in modulating glucose metabolism and oxidative stress pathways in diabetes management [65].

While these findings are promising, further research, particularly in human subjects, is necessary to fully understand the extent and mechanisms by which chlorophyll and derivatives can influence glucose absorption and metabolism.

## 7. The Role of Chlorophyllin

Chlorophyllin is a semi-synthetic water-soluble derivative of chlorophyll, often used in research and supplements due to its stability and solubility [66,67]. Chlorophyllin is produced through alkaline hydrolysis of chlorophyll, with replacement of the magnesium ion, most commonly with copper or zinc, absence of the phytyl tail, and presenting carboxyl groups—from hydrolysis of chlorophyll’s fifth ring as in chlorin e6—on the chlorin ring [28,66,68] (Figure 7). Chlorophyllin can interact with environmental toxins and has a history of safe medical use [66,67]. Approved by the FDA for internal deodorant use at doses up to 300 mg/day, it also helps manage odours in elderly patients and supports wound healing. Additionally, chlorophyllin serves as a food additive and colouring agent. Research has shown that it has antioxidant and antigenotoxic effects, and it has recently gained attention for its potential anticancer properties [66,67].

Chlorophyllin has been shown to acutely lower postprandial blood glucose levels in humans, as observed in a small-scale glucose tolerance test [69]. While the precise mechanism remains unclear, it has been hypothesized that this effect may involve modulation of intestinal glucose transporters or incretin signalling, possibly acting as a GLP-1 secretagogue [69]. Supporting this hypothesis, several clinical studies have demonstrated that daily supplementation with green plant membranes (thylakoids), which are naturally rich in chlorophyll, can enhance GLP-1 secretion, reduce hunger, and reduce obesity-related risk factors. For example, Montelius et al. [70] reported that a three-month supplementation with green plant membranes in overweight women led to increased GLP-1 levels, body weight loss, and reduced cravings for palatable food. Similarly, Stenblom et al. [71] demonstrated that thylakoid intake prior to a carbohydrate-rich meal significantly increased GLP-1 and cholecystokinin (CCK) levels while reducing postprandial hypoglycaemia and hunger. These findings were supported by a systematic review by Amirinejad et al. [72], which concluded that thylakoids enhance satiety and support weight loss, likely through incretin-related mechanisms. This effect of chlorophyll and its derivatives may be attributed to their proposed DPP-4 inhibitory activity, resulting in elevated incretin levels, including GLP-1, as above discussed referring to the work by Turkiewicz et al. [43]. Together, these findings support a broader role for chlorophyll-rich plant components in appetite regulation and glycaemic control, although further studies are needed to isolate the specific contribution of chlorophyll or derivatives and clarify its direct mechanistic role in GLP-1 signalling.

Chlorophyllin has demonstrated significant potential in mitigating diabetes-induced hepatic damage. In a study involving streptozotocin-induced diabetic mice [73], sodium–copper chlorophyllin supplementation resulted in a marked reduction of oxidative stress markers, including reactive oxygen species (ROS), malondialdehyde (MDA), and protein carbonyl levels in the liver. Concurrently, there was an upregulation of the activity and expression of key antioxidant enzymes such as copper–zinc superoxide dismutase (CuZnSOD), manganese superoxide dismutase (MnSOD), and catalase, indicating an enhanced antioxidant defence mechanism. Furthermore, chlorophyllin exhibited hepatoprotective effects, as evidenced by decreased activities of liver enzymes like aspartate aminotransferase (AST), alanine aminotransferase (ALT), and alkaline phosphatase (ALP). Histological analyses corroborated these findings, revealing that chlorophyllin ameliorated morphological and cellular alterations in the liver tissue of diabetic mice. At the molecular level, chlorophyllin modulated apoptotic pathways by downregulating pro-apoptotic proteins such as caspase-3 and caspase-9, while upregulating the anti-apoptotic protein B-cell lymphoma-2 (Bcl-2). These multifaceted actions suggest that chlorophyllin exerts its protective effects through the modulation of hyperglycaemia-induced oxidative stress and apoptosis, highlighting its potential as a therapeutic agent in the management of diabetes-related hepatic complications. The same authors [74] had also demonstrated the therapeutic potential of chlorophyllin in managing diabetes, specifically its effects on hyperglycaemia and hyperlipidaemia, using the same streptozotocin-induced diabetic mouse model as in the previous study. Chlorophyllin, at the optimum dose of 50 mg/kg body weight, significantly reduced blood glucose levels in both acute and sub-acute treatment periods. In an intraperitoneal glucose tolerance test (IPGTT), diabetic mice treated with chlorophyllin showed marked reductions in glucose levels at both 2 and 4 h compared to untreated diabetic controls. Furthermore, chlorophyllin treatment led to a significant reduction in glycosylated haemoglobin levels and normalized lipid profiles, indicating not only antihyperglycaemic but also antihyperlipidaemic effects.

Zheng et al. [75] investigated the effects of chlorophyllin on liver fibrosis induced by treatment with carbon tetrachloride in BALB/c mice. While the study does not directly address diabetes, its findings are relevant to metabolic conditions often associated with diabetes. The researchers found that oral administration of chlorophyllin attenuated both intestinal and hepatic inflammation and ameliorated liver fibrosis. Importantly, chlorophyllin treatment strengthened intestinal barrier integrity, reducing intestinal permeability and restoring tight junction proteins in the ileum; additionally, the treatment rebalanced the gut microbiota, characterized by a decrease in the phylum Firmicutes and an increase in Bacteroidetes. Additionally, in vitro experiments demonstrated that chlorophyllin inhibited the NF-κB pathway via suppression of IKK phosphorylation, suggesting a mechanism for its anti-inflammatory effects. These findings suggest that chlorophyllin may have potential applications in regulating intestinal microbiota and reducing inflammation, which are factors relevant to metabolic diseases such as diabetes. A more robust intestinal barrier can prevent systemic inflammation and endotoxaemia, factors that are known to impair insulin sensitivity and glucose metabolism [75].

A study by Samadder et al. [76] investigated the protective effects of chlorophyllin on glucose metabolism disrupted by alloxan, which induces diabetes and mitochondrial dysfunction. In a mouse model, pre-treatment with chlorophyllin significantly delayed the onset of diabetes and mitigated the alloxan-induced rise in blood glucose levels. Chlorophyllin helped restore glucose metabolism by optimizing key metabolic and enzymatic markers (such as glucokinase, pyruvate, and insulin), while also modulating the expression of insulin signalling proteins (IRS1, IRS2, and GLUT2). Additionally, chlorophyllin stabilized mitochondrial function by maintaining the membrane potential, ATP/ADP ratio, and ATPase activity, thus preserving energy production and reducing cellular damage.

## 8. Chlorophyll-Based Compounds with Antidiabetic Potential

There is growing scientific interest in chlorophyll-based compounds as promising candidates for future development in diabetes treatment strategies. In this context, as discussed above, pheophorbide *a* has demonstrated notable antidiabetic potential. In a study by Ribeiro et al. [77], the authors conducted a phenotypic screening of 182 fractions derived from 19 cyanobacterial strains using zebrafish larvae, aiming to identify compounds with insulin-mimetic activity. Two fractions, named 06104_D and 03283_B, were found to significantly enhance glucose uptake, as demonstrated by the 2-NBDG fluorescent glucose analogue assay. Further analysis revealed that fraction 06104_D not only lowered free glucose levels but also upregulated the expression of the insulin gene *insa* in zebrafish larvae. Metabolomic profiling of this active fraction identified several chlorophyll derivatives, with pheophorbide *a* emerging as a major constituent. Subsequent testing of commercially available pheophorbide *a* confirmed its ability to stimulate glucose uptake in vivo, highlighting its potential as a novel insulin-mimetic compound.

These findings align with previous reports supporting the antidiabetic potential of chlorophyll derivatives, also derived from sources of ethnobotanical interest. Attention has been drawn to *Clerodendrum infortunatum* L., a shrub from the Verbenaceae family that is widely distributed across tropical and subtropical regions and is traditionally known as “Hill Glory Bower” in English and “Bhant” in Bangla; it is described to hold ethnomedicinal importance among tribal communities in Bangladesh [78]. Recent studies have indicated that the leaves of *Clerodendrum infortunatum* exhibit anti-diabetic activity [79]. However, the specific bioactive compound (s) responsible for this effect remained to be identified, until research was directed to evaluate the anti-diabetic potential of its ethanolic leaf extracts and to isolate and characterize the active constituents responsible for this activity. From the leaves of *Clerodendrum infortunatum,* pheophytin *a* was isolated as fraction-1 and has been shown to reduce both blood glucose and cholesterol levels in rats, accompanied by increased insulin secretion following oral administration (1.25 g/kg body weight), suggesting a significant anti-diabetic action [78].

Moreover, pheophorbide *a* has demonstrated significant inhibitory activity against the receptor for advanced glycation end products (RAGE), a key mediator in the development of diabetic complications. In a study by Matsumoto et al. [80], ethanol extracts from the leaves of *Mallotus japonicus*, a plant that is used in the Japanese pharmacopoeia against gastric diseases, were found to inhibit the binding of AGEs to RAGE. Activity-guided fractionation and LC/MS analysis identified pheophorbide *a* as a principal component responsible for this activity. Pheophorbide *a* demonstrated strong inhibition of the AGE–RAGE interaction, with an IC_50_ of 0.102 μM, closely matching the efficacy of the established RAGE inhibitor dalteparin, which has an IC_50_ of 0.084 μM. The authors proposed that pheophorbide *a* holds potential as a natural compound for development as a therapeutic or nutraceutical agent targeting health issues associated with AGE–RAGE axis activation.

Paul et al. [81] investigated the role of pheophorbide *a* as an insulin mimetic, focusing on its interaction with glucose transporters (GLUTs), in particular GLUT1, which is expressed in many cell types, including pancreatic β-cells, and permits basal glucose uptake, and GLUT4, the insulin-responsive transporter of glucose in peripheral tissues [82,83]. GLUT1 is required for the activity of pancreatic β-cells leading to insulin secretion [83]. Using in silico molecular docking, pheophorbide *a* demonstrated a strong binding affinity to GLUT1 and GLUT4 transporters, surpassing that of metformin and glucose [81]. Molecular docking and 100 ns molecular dynamics simulations revealed that pheophorbide *a* forms a highly stable complex with GLUT1, consistent hydrogen bonding, minimal structural fluctuations and sustained structural compactness, highlighting its strong binding affinity and potential as a GLUT1-targeted therapeutic agent in diabetes [81]. Complementing these computational findings, in vitro experiments with INS-1 cells (rat insulinoma cell line secreting insulin) showed that pheophorbide *a* treatment increased GLUT1 density at the plasma membrane under high glucose conditions, enhanced glucose uptake, and reduced GLUT1 mobility, suggesting a stabilized presence of the transporter [81]. These findings indicate that pheophorbide *a* could facilitate glucose-stimulated insulin secretion by modulating GLUT1 trafficking and function, positioning it as a potential candidate for diabetes management strategies. Paul et al. [81] also evaluated the potential of pheophorbide *a* as a therapeutic agent in diabetes, with particular attention to its pharmacokinetic and drug-likeness profiles. Computational ADMET (absorption, distribution, metabolism, excretion, and toxicity) predictions suggested that the compound is generally non-toxic, exhibiting no predicted mutagenic, carcinogenic, immunotoxic, or hepatotoxic effects, and a relatively safe toxicity classification [81]. Its physicochemical properties, including a moderate lipophilicity (cLogP 3.82) and a topological polar surface area (133 Å^2^), are consistent with acceptable membrane permeability, supporting its suitability for drug development. Although the molecular weight of 592.7 slightly exceeds the conventional limit of 500 Da, it maintains compliance with several criteria in Lipinski’s Rule of Five. Pheophorbide *a* shows limited gastrointestinal absorption and poor blood–brain barrier permeability, which may be advantageous for targeting peripheral tissues without central nervous system effects. Moreover, it is important to note that the inhibitory activity of pheophorbide *a* on intestinal α-glucosidase and α-amylase, as discussed above, does not require systemic absorption; like acarbose, this action is exerted locally within the intestinal lumen, which may mitigate concerns regarding its bioavailability for this specific mechanism of action. However, to direct the compound into specific sites of cellular action, such as pancreatic β-cells, it might be interesting to find strategies to overcome pheophorbide *a*’s pharmacokinetic limitations, particularly its high molecular weight and poor gastrointestinal absorption. Recent research has focused on innovative delivery systems that can improve its solubility, targeting, and bioefficacy. One approach includes PLGA-based nanoparticles modified with polyethylene glycol and folate for active tumour targeting; these folate–poly(lactic-co-glycolic acid) nanoparticles demonstrated enhanced pheophorbide *a* uptake and retention in gastric cancer models (MKN28) in vivo, suggesting improved delivery efficiency [84]. Another promising strategy utilizes amphiphilic copolymer nanoparticles formed from N-vinylpyrrolidone, which encapsulate pheophorbide (or its chlorin derivative) and significantly enhance its photophysical activity upon interaction with biomembranes, indicating better bioavailability in situ [85]. Additionally, casein micelles, naturally evolved carriers for hydrophobic compounds, have been shown to stably encapsulate pheophorbide *a* (final micelle size was about 220 nm) and facilitate efficient cellular uptake and photodynamic activity in tumour cells [86]. Finally, poloxamer F127 (a non-ionic copolymer surfactant) micelles co-loaded with doxorubicin and pheophorbide *a* have been developed for combined chemo–photodynamic therapy; these micelles maintained structural integrity and demonstrated strong tumour growth inhibition in vivo upon light activation [87]. Although originally developed for cancer applications, these diverse delivery platforms collectively show promise for enhancing the clinical translation of pheophorbide *a* by addressing its solubility, bioavailability, and targeting challenges. This suggests that pheophorbide *a* may also hold potential as an orally deliverable agent for modulating metabolic dysfunctions in diabetes.

Interestingly, Paul et al. [88], in another investigation, had earlier reported that the activity of a commercial antidiabetic polyherbal formulation from Indian markets (containing *Mangifera indica*, *Momordica charantia, Gymnema sylvestre* and *Syzygium cumini*), can be attributed to the presence of pheophorbide *a* and pyropheophorbide *a* (decarbomethoxylated pheophorbide *a*), identified in a hydro-alcoholic extract by using HPLC–ESI–MS/MS. The previously discussed risk of phototoxic skin reactions resembling pseudoporphyria, associated with pheophorbide *a* exposure, must always be carefully considered [51,52,53,54]. While the therapeutic potential of pheophorbide *a* in metabolic disorders such as diabetes is an emerging area of interest, it is essential to also acknowledge its well-documented phototoxic properties, particularly from studies in oncology [89,90,91,92,93]. Upon light activation, pheophorbide *a* undergoes photochemical reactions that produce reactive oxygen species, including singlet oxygen, hydroxyl radicals, and superoxide anions, which damage cellular components and lead to cell death [89]. As a potent photosensitizer, pheophorbide *a* has demonstrated significant efficacy in preclinical cancer models through photodynamic mechanisms; however, this same activity raises concerns regarding photosensitivity and unintended tissue damage upon light exposure. Several in vivo and in vitro cancer studies have reported cutaneous phototoxicity and systemic oxidative stress following light activation of pheophorbide *a*, even at low doses [94]. Although such effects are context-dependent and typically arise under photoactivating conditions, this inherent characteristic underscores the need for caution when translating pheophorbide *a* to non-oncological settings. In the context of diabetes management, where chronic administration may be considered, minimizing exposure to light or modifying the molecular structure to reduce its photosensitizing potential should be carefully evaluated in future preclinical and clinical investigations.

Taken together, these data indicate the therapeutic potential of chlorophyll-derived compounds as possible candidates for the development of antidiabetic agents targeting multiple aspects of glucose regulation, in particular glucose transport pathways, besides the aforementioned activity on carbohydrate digestion.

## 9. Chlorophylls and Pheophytins as Antioxidants

Chlorophylls and their derivatives have attracted increasing interest due to their complex biological roles, which include both antioxidant and pro-oxidant activities. These pigments may contribute to cancer prevention not only by neutralizing reactive species and modulating detoxification pathways but also by promoting selective oxidative stress and apoptosis in tumour cells [95]. The complete loss of cytotoxic activity in dechlorophyllized olive leaf and green tea extracts highlights the essential contribution of chlorophylls to their anticancer effects, suggesting that these pigments are key bioactive components likely responsible for inducing tumour cell death, independent of their phenolic content [95].

The properties linked to their antioxidant effects are attracting increasing attention from scientific and industrial communities [21,34,96]. As discussed in the previous sections, recent findings highlight the potential of chlorophyll-derived compounds to modulate glucose homeostasis. Notably, pheophorbide *a*, identified through phenotypic screening in zebrafish larvae, demonstrated insulin-mimetic activity by enhancing glucose uptake and reducing glucose levels in vivo [77]. While this metabolic activity is central to their therapeutic promise, these compounds also exhibit well-documented antioxidant properties, which may offer complementary benefits. For instance, chlorophyll *a* and *b*, as well as pheophytin *a* and *b*, which are abundant in avocado oil [97], have been recognized for their antioxidative capacity [98], potentially contributing to broader cellular protection and metabolic regulation. These compounds may mitigate hyperglycaemia-induced oxidative stress, a major factor in the deterioration of β-cell function and the development of insulin resistance. Moreover, degradation products of chlorophyll *a*, such as pheophorbide *a* and chlorophyllin, have been shown to inhibit NF-κB activation, a key transcription factor regulating pro-inflammatory cytokine expression, suggesting an additional anti-inflammatory mechanism relevant to diabetes pathophysiology [75,99]. By reducing oxidative stress and inflammatory signalling, chlorophyll derivatives could help preserve insulin signalling pathways and promote metabolic homeostasis, thereby offering a multifaceted therapeutic approach for diabetes management.

## 10. Conclusions

In conclusion, the current body of evidence highlights the promising role of chlorophyll and its derivatives as multifaceted antidiabetic agents (Figure 8). As illustrated, these mechanisms include digestive enzyme inhibition, incretin modulation, nuclear receptor activation, glucose transporter regulation, and attenuation of oxidative stress pathways. While preclinical studies using zebrafish, rodent, and in silico models provide valuable mechanistic insights, they provide limited predictive value for human efficacy and safety. On average, only about 5% of therapies tested in animal models ultimately gain regulatory approval for human use, underscoring the limitations of preclinical approaches [100], a consideration that applies directly to the present context. Moreover, the pharmacokinetic profiles of many chlorophyll derivatives remain poorly characterized, complicating assessments of their bioavailability, metabolism, and dose optimization in human systems. Safety concerns also warrant closer attention. For example, the phototoxic potential of pheophorbide *a*, a known chlorophyll degradation product, has been documented under light exposure and may pose a significant risk in therapeutic applications. These factors underscore the need for caution when evaluating the clinical readiness of these compounds. To repurpose pheophorbide *a* as an intestinal α-glucosidase inhibitor for diabetes, strategies such as chemical derivatization to reduce singlet-oxygen yield, conjugation to non-absorbable carriers, or formulation in gut-targeted delivery systems could reduce the undesired photochemical reactivity and systemic distribution to minimize the phototoxic risk while preserving local enzymatic inhibition.

Recent analyses further highlight these translational gaps and advocate for more human-relevant methodologies to improve predictive accuracy [101]. Our literature search did not identify any published human clinical trials specifically evaluating chlorophyll or its derivatives for diabetes management, highlighting a critical unmet need. To help address this, innovative strategies such as organ-on-a-chip platforms, human tissue-based assays, and in silico clinical trial simulations could be explored as more predictive alternatives [100]. Additionally, systematic reviews and meta-analyses of animal studies may help identify the most promising therapeutic candidates and guide the design of future clinical trials. Specific next steps could include target validation in human-relevant systems, formulation optimization, and dose–response studies to support safe and effective translation to clinical contexts. Continued research, particularly in advanced preclinical and clinical settings, is essential to determine the true therapeutic potential and safety of chlorophyll-based interventions in diabetes management.

## Figures and Tables

**Figure 1 nutrients-17-02653-f001:**
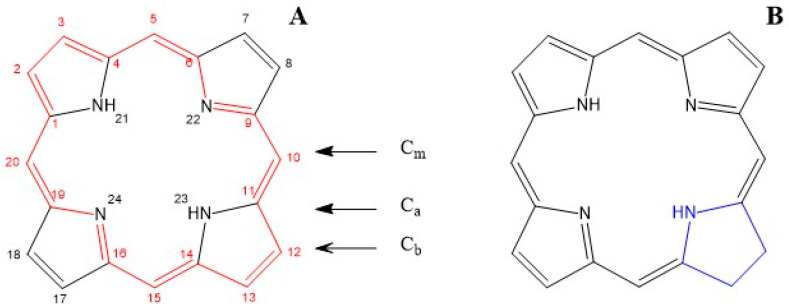
Porphyrin (**A**) with the IUPAC atom numbering and carbon types, and the chlorin ring (2,3-dihydroporphyrin, (**B**)) with the hydrogenated site in blue.

**Figure 2 nutrients-17-02653-f002:**
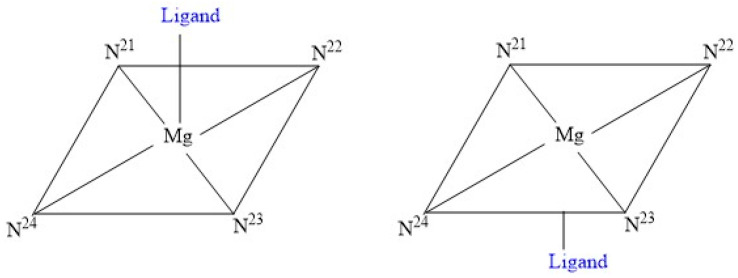
Schematic drawing for single axial coordination of chlorophylls to form five-coordinated species with chirality at the central magnesium.

**Figure 3 nutrients-17-02653-f003:**
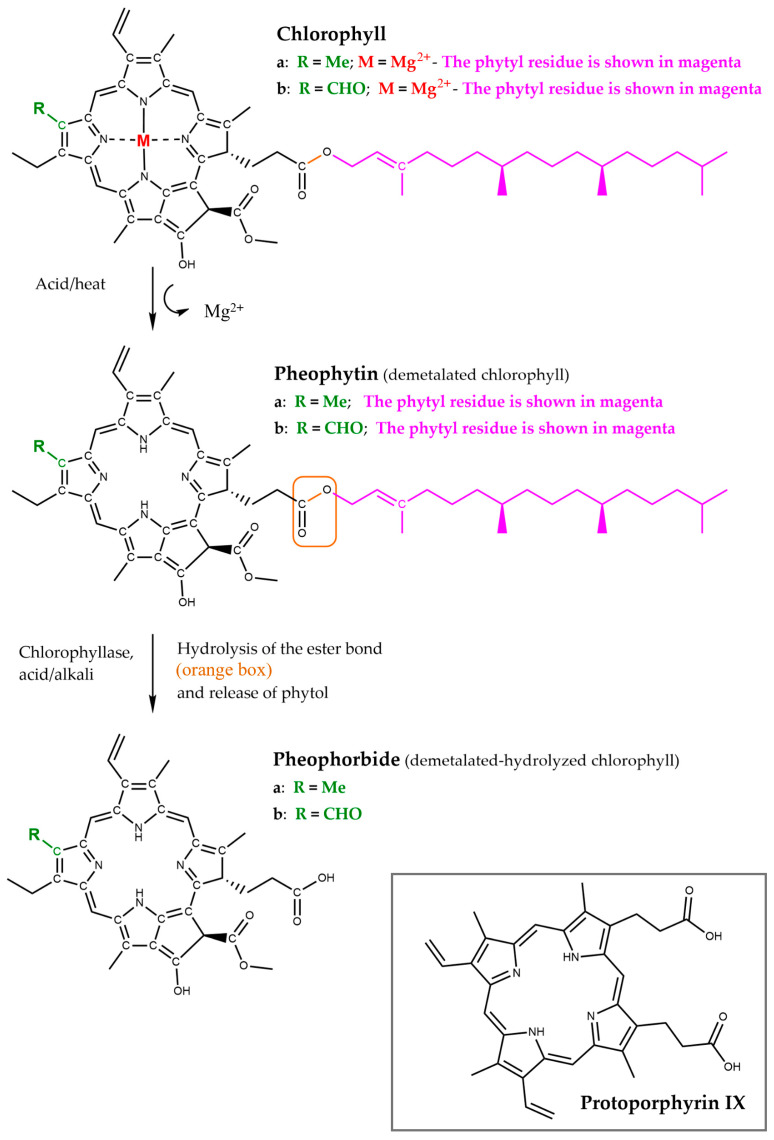
Chemical structure of the most abundant chlorophylls—chlorophyll *a* and chlorophyll *b*—and of the degradation products phaeophytin and pheophorbide. Inset: protoporphyrin IX.

**Figure 4 nutrients-17-02653-f004:**
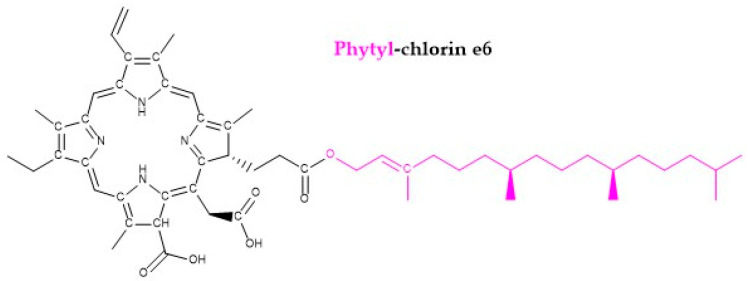
Chemical structure of the phytyl-chlorin e6, according to [36]. The phytyl residue is shown in magenta.

**Figure 5 nutrients-17-02653-f005:**
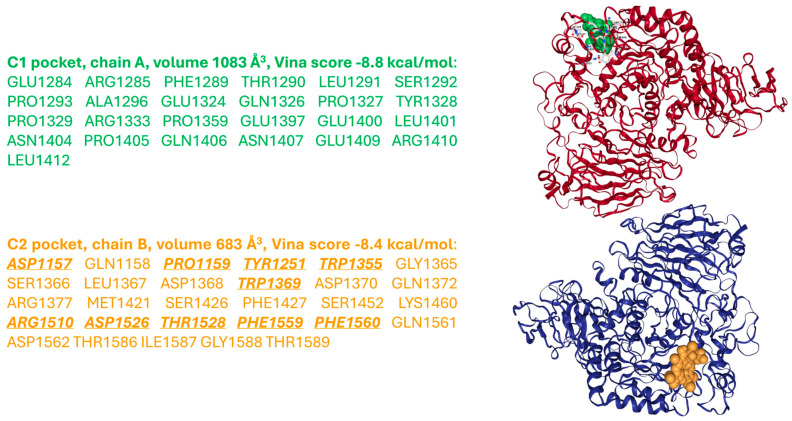
Molecular docking obtained with CB-Dock2 web-based software showing the most relevant interactions of pheophorbide *a* (shown as green and orange space filling molecules) with the human maltase-glucoamylase enzyme (PDB ID: 3TOP). In the C2 pocket, the residues of the active site, identified also for acarbose interactions [47], are in bold and underlined. Protein structures of chains A and B are shown as ribbon diagrams in red and blue, respectively.

**Figure 6 nutrients-17-02653-f006:**
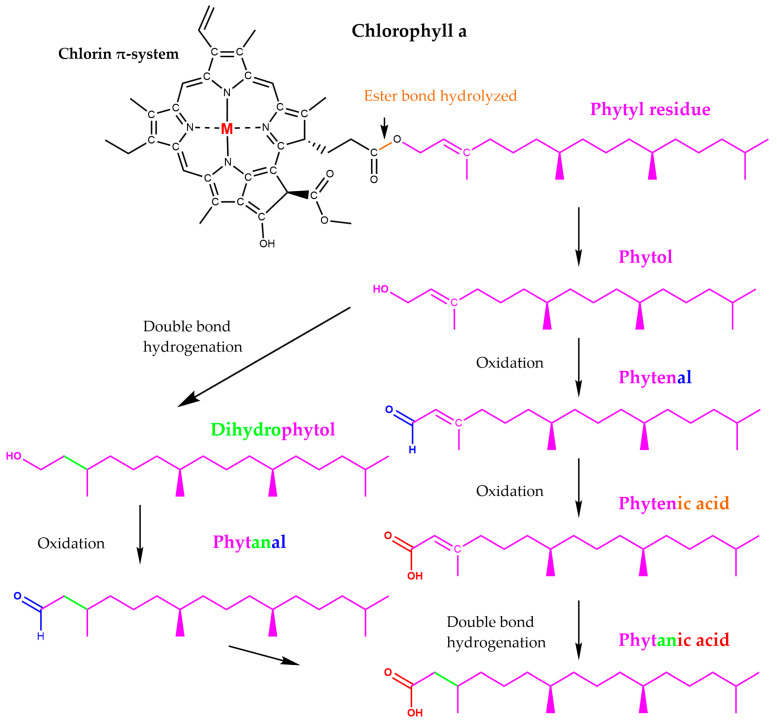
Phytanic acid from chlorophyll. Drawn based on chemical structures presented in reference [56]. Colour coding is used for visual distinction between different functional groups and molecular modifications.

**Figure 7 nutrients-17-02653-f007:**
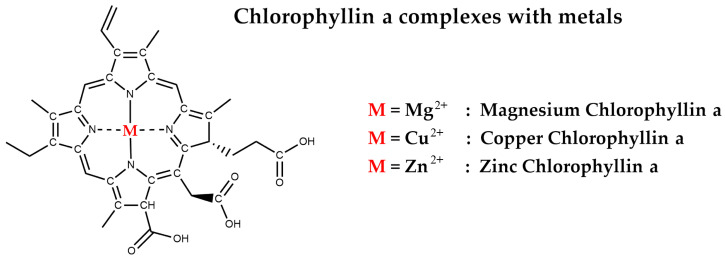
Magnesium, copper and zinc chlorophyllin *a*.

**Figure 8 nutrients-17-02653-f008:**
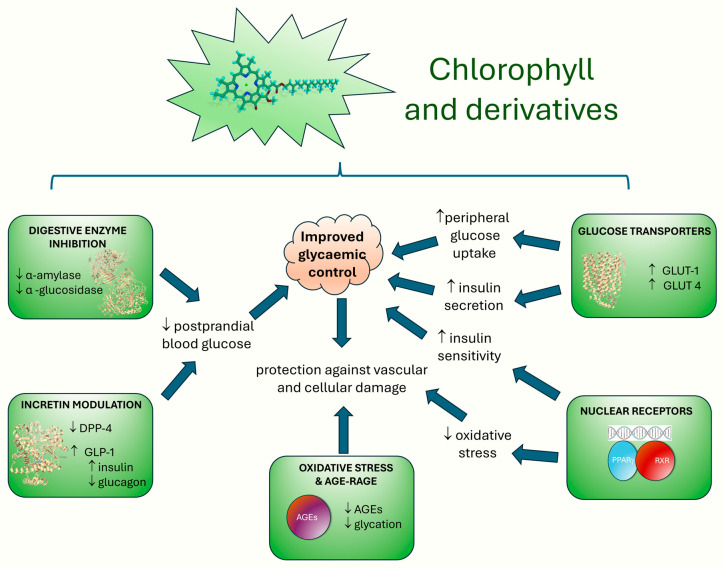
Proposed mechanistic pathways by which chlorophyll and its derivatives may exert antidiabetic effects. This schematic summarizes the multifaceted biological actions identified across preclinical studies.

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
