# Peer review of "Beyond Green: The Therapeutic Potential of Chlorophyll and Its Derivatives in Diabetes Control"

_nutrients, 2025, doi:10.3390/nu17162653_

Round 1
Reviewer 1 Report
Comments and Suggestions for Authors
This review explores an emerging area of metabolic research: the potential therapeutic role of chlorophyll and its derivatives in diabetes management. While the premises are timely and the evidence intriguing, several important strengths and limitations should be critically addressed.
Although the review draws from preclinical studies, it fails to cite or discuss any significant human clinical trials. This is a major gap, as the translation from animal or in vitro models to human outcomes is often inconsistent. Without clinical validation, the therapeutic relevance remains speculative.
As a narrative review, the article does not clarify how studies were selected, evaluated, or synthesized. Without a clear methodological framework (e.g., PRISMA or systematic criteria), the risk of selection bias and cherry-picking of positive outcomes increases.
Nearly all supporting evidence comes from zebrafish, rodent, or in silico models. While promising, these cannot predict human efficacy or safety. The lack of clinical or even advanced preclinical (e.g., large animal) data weakens the therapeutic claim.
The pharmacokinetic profile of pheophorbide a, including its high molecular weight and poor GI absorption, raises questions about its bioavailability and practical formulation. These challenges are acknowledged but downplayed, while drug delivery solutions or alternatives (e.g., nanoparticle encapsulation, transdermal systems) are not discussed.
The conclusion succinctly summarizes the central message of the chapter by acknowledging the promising antidiabetic potential of chlorophyll and its derivatives, especially in modulating glucose transport and carbohydrate digestion. It effectively reinforces the multifaceted nature of these compounds and their potential therapeutic relevance. However, while it accurately identifies the need for further clinical investigation, it remains somewhat generic and overly optimistic, lacking a nuanced discussion of key limitations.
For example, the preclinical nature of most cited evidence, reliance on in silico models, and the limited pharmacokinetic data should be explicitly emphasized to contextualize the current readiness of chlorophyll derivatives for drug development. In addition, safety concerns, particularly the documented risk of phototoxicity associated with pheophorbide are notably absent from the final statement. Ignoring such risks may give the impression of unjustified confidence in therapeutic translation.
Finally, while the call for clinical studies is appropriate, it would be strengthened by suggesting specific next steps, such as target validation, formulation studies, or dose-response trials, to underscore the path forward from bench to bedside.
Reviewer 2 Report
Comments and Suggestions for Authors
The article entitled Beyond Green: The Therapeutic Potential of Chlorophyll and its Derivatives in Diabetes Control provides a valuable and comprehensive summary of current data on the multidirectional effects of chlorophyll and its derivatives in the context of diabetes therapy. The authors present the complex mechanisms of action of these compounds on glucose metabolism, digestive enzymes, and signaling pathways in a clear and substantive manner. The review is well documented in the literature and is interdisciplinary in nature, making it a valuable contribution to the development of natural strategies to support diabetes treatment. Below are a few comments to improve the work:
1. The authors present numerous data from in vitro and in vivo studies (on animal models) in their review, but conclusions regarding the potential therapeutic application of chlorophyll derivatives in the treatment of diabetes require a clear emphasis on the need for further clinical studies.
I suggest expanding the Conclusions section to clearly indicate the translational limitations of the current data.
2. Lines 248-260: The article refers to the inhibitory activity of microencapsulated chlorophyll extracts against DPP-4, but there is a lack of detailed information on the molecular mechanism and the strength of binding to the active sites of the enzyme. I would ask the authors to supplement this section with more precise data (eg. IC50 values).
3. Lines 382-385: Although the paper mentions the possibility of chlorophyllin acting as a GLP-1 secretagogue, there is no detailed explanation of the mechanism and no reference to research results confirming this effect. Please supplement this part of the paper with data from the literature or clearly indicate that these are only hypotheses requiring further research.
4. Lines 513-515: The article mentions the risk of phototoxicity of pheophorbide a and the phenomenon of pseudoporphyria, but this topic is only marginally addressed in the context of clinical safety. I recommend adding a subchapter or expanding the discussion on the limitations and risks associated with chlorophyll derivative supplementation (in the context of possible implementation in diabetes therapy).
5. Due to the complexity of the subject (multiple mechanisms of action of chlorophyll and its derivatives), it would be useful to include a simplified graphic diagram (mechanistic pathway) showing the interaction with digestive enzymes (α-amylase, α-glucosidase, DPP-4), glucose transporters (GLUT1, GLUT4), RXR/PPARγ modulation, and the impact on the AGE–RAGE axis and oxidative stress.
Round 2
Reviewer 1 Report
Comments and Suggestions for Authors
All comments addressed satisfactory.